# De Novo Hybrid Assembly Unveils Multi-Chromosomal Mitochondrial Genomes in *Ludwigia* Species, Highlighting Genomic Recombination, Gene Transfer, and RNA Editing Events

**DOI:** 10.3390/ijms25137283

**Published:** 2024-07-02

**Authors:** Guillaume Doré, Dominique Barloy, Frédérique Barloy-Hubler

**Affiliations:** 1DECOD (Ecosystem Dynamics and Sustainability), Institut Agro, INRAE, IFREMER, 35042 Rennes, France; guillaume.dore@agrocampus-ouest.fr; 2UMR 6553 ECOBIO, CNRS, Université de Rennes 1, 35042 Rennes, France

**Keywords:** water primrose, Myrtales, mitogenome, hybrid assemblies, DNA transfer, repeated sequences

## Abstract

Biological invasions have been identified as the fifth cause of biodiversity loss, and their subsequent dispersal represents a major ecological challenge. The aquatic invasive species *Ludwigia grandiflora* subsp. *hexapetala* (*Lgh*) and *Ludwigia peploides* subsp. *montevidensis* (*Lpm*) are largely distributed in aquatic environments in North America and in Europe. However, they also present worrying terrestrial forms that are able to colonize wet meadows. To comprehend the mechanisms of the terrestrial adaptation of *Lgh* and *Lpm*, it is necessary to develop their genomic resources, which are currently poorly documented. We performed de novo assembly of the mitogenomes of *Lgh* and *Lpm* through hybrid assemblies, combining short reads (SR) and/or long reads (LR) before annotating both mitogenomes. We successfully assembled the mitogenomes of *Lgh* and *Lpm* into two circular molecules each, resulting in a combined total length of 711,578 bp and 722,518 bp, respectively. Notably, both the *Lgh* and *Lpm* molecules contained plastome-origin sequences, comprising 7.8% of the mitochondrial genome length. Additionally, we identified recombinations that were mediated by large repeats, suggesting the presence of multiple alternative conformations. In conclusion, our study presents the first high-quality mitogenomes of *Lpm* and *Lgh*, which are the only ones in the Myrtales order found as two circular molecules.

## 1. Introduction

### 1.1. Origin, Biological Traits, and Distribution of the Aquatic Invasive Plants, Ludwigia Species

Biological invasions are now recognized as one of the primary drivers behind global biodiversity loss. Invasive alien plant species (IAPs) lead to significant economic and ecological losses by causing disruption to ecosystems [1]. IAPs are characterized by their strong growth, excellent competitiveness, high adaptive value, an absence of predators, and also reproduce abundantly [2,3].

Of all these invasive aquatic plant species, *Ludwigia grandiflora* subsp. *hexapetala* (or *Lgh*) and *Ludwigia peploides* subsp. *montevidensis* (or *Lpm*) stand out in particular. They’re native to South America, but both species have colonized many countries around the world [4]. In North America, *Lgh* and *Lpm* are distributed across several states where they have both been found to degrade major watersheds as well as aquatic and riparian ecosystems [5]. *Lpm* and *Lgh* show some advantageous biological traits that might explain their success of colonization in news areas. *Lpm* and *Lgh* reproduce essentially by clonal propagation with high grow rate, which contribute to the dispersal and establishment of propagules [6]. Both *Ludwigia* species are capable of thriving in a wide range of environments, revealing their broad ecological tolerance and great plasticity [7]. Sexual production is also effective in both *Ludwigia* species. In France, *Lpm* is self-compatible and produces many capsules and seeds [8]. *Lgh*, on the other hand, possesses a heteromorphic late-acting self-incompatible system and is also able to produce seeds [9]. Seeds can be dispersed by floating fruit, which contributes to long-distance colonization, as is the case for propagules [7]. *Lgh* has provoked similar damage in Europe, and has been observed in France [10], Germany [11], Italy [12], Spain [13], Switzerland [14], Great Britain/Ireland [15], and Turkey [16]. *Lgh* is also present in Japan [17]. *Lpm* is very common in the southeast and west of France, but its distribution is more limited in Belgium, Corsica, Greece, Italy, the Netherlands, Spain, Turkey, the Balkans, and the UK [18,19]. The oldest documented introductions for both species were in France in 1830 [20].

In its natural habitat, both *Ludwigia* ssp. exhibit an amphibious nature [21], and are commonly found in wetland areas or in transitional zones between aquatic and terrestrial environments [22,23]. However, it is now developing particularly worrying terrestrial forms and is able to colonize wet meadows at an alarming rate [24]. *Lgh* has two distinct morphological types (morphotypes) depending on its environment (aquatic or terrestrial). The two morphotypes present different metabolic processes: fermentation for the aquatic morphotype, and respiration for the terrestrial morphotype [25].

### 1.2. Genomic Resources in the Onagraceae Family

Genomics plays a central role in identifying genes and metabolic pathways involved in the adaptation process leading to plant invasion [3], with genomic information helping to predict and control invasiveness [26]. For instance, in species such as *Reynoutria japonica* and *Phalaris arundinacea*, genes that play a critical role in invasion are associated with stress response, reproduction, and growth [27,28]. Hence, the genomics of invasive plants represents a powerful tool for comprehending the molecular foundations of plant invasion, enabling the prediction and management of their proliferation [29,30].

*Lgh* and *Lpm* belong to the malvids clade, Myrtales order, Onagraceae family, and Ludwigioideae subfamily. To date, *Ludwigia* species have not undergone comprehensive genome sequencing, with the exception of the chloroplast genomes of *L. octovalvis* [31], *Lgh* and *Lpm* [32]. In the related subfamily Onagroideae, two incomplete draft genomes are available for *Chamaenerion angustifolium* and *Oenothera biennis* (unpublished GenBank records). *Lpm*, one of the progenitor species of *Lgh*, is a diploid species (2n = 16) with a relatively small nuclear genome (261.7 Mb), and *Lgh* is a decaploid (2n = 80) with a very large nuclear genome (1.4 Gb) [33]. Assembly of the *Lgh* genome is anticipated to be challenging and time-consuming, as has been acknowledged in previous studies dealing with such big and complex genomes [34]. Having successfully assembled and annotated their plastomes [32], we decided to subsequently assemble their mitogenomes due to the crucial role they play in the plants’ biological functions [35,36]. In addition, no Ludwigioideae mitogenomes are currently available, and only four Onagraceae mitogenomes: *Chamaenerion angustifolium* (OX328283) and *Oenothera elata*, *O. biennis*, and *O. villaricae* [37].

### 1.3. Form and Size of Plant Mitogenomes

Commonly depicted as circular, mitogenomes of terrestrial plants exhibit intricate, diverse, and dynamic structures. Hence, certain plants possess multipartite mitochondrial genomes consisting of several chromosomes, predominantly in a circular form, although they can be linear or branched [38,39,40,41]. The most commonly reported structure consists of two circular mitochondrial chromosomes, as demonstrated by numerous studies for *Allium cepa* [42], *Saccharum officinarum* [43], *Fallopia multiflora* [44], *Salvia officinalis* [45], and *Zantedeschia odorata* [46]. Another common structure consists of three circular molecules like *Cynanchum wilfordii* [47], *Brassica campestris* [48], and cucumber [49]. Recent studies have revealed the existence of even more intricate mitochondrial genomes. For instance, *Amborella trichopoda* has five circular chromosomes [50], *P. micranthum* has 26 circular subgenomes [51], *Lophophytum mirabile* is divided into 54 circular chromosomes [52], *Silene noctiflora* has 59 to 63 circular chromosomes [53], and *Silene conica* has at least 128 circular chromosomes, some of which are empty (i.e., contain no functional genes) [54].

In addition to these structural variations, angiosperm mitochondrial genomes are particularly variable in size. While the mitogenome of *Viscum scurruloideum* is frequently referenced in articles as being the smallest, measuring 66 kb [55], it is worth noting that smaller unpublished mitogenomes exist in the GenBank database including that of Asteraceae *Carthamus tinctorius* (63 kb, OQ621746) or Orchidaceae *Dendrobium amplum* (48 kb, MH591889). In contrast, the largest mitogenome currently described is that of *Larix sibirica* with a total of 11.7 Mb [56].

### 1.4. Repeat Sequences and Exogenous DNA in Plant Mitogenomes

Vascular plant mitogenomes also contain a large fraction of repeat sequences, making up between five and ten percent of the genome [57]. Repetitive elements, especially within the Angiosperm clade, can reach significant sizes, with repeats exceeding 10 kb being relatively common [58]. A portion of these repeated sequences can be traced back to transposable elements such as retrotransposons and mitoviruses [59,60], and numerous TEs fragments have been reported in plant mitogenomes [61,62,63]. The presence of these extensive repeated sequences within mitogenomes has made it difficult to assemble them accurately [57,64]. Repeats, especially those exceeding 1000 bp, play a significant role in the homologous recombination phenomena [65,66], resulting in different conformations coexisting within the same species [67,68].

The ability of the mitochondrial genome to incorporate exogenous DNA also partly explains these size variations [69]. The most frequent transfer concerns the insertion of chloroplast DNA sequences [70], although sequences of nuclear origin can also be found inserted [71]. Nucleotide sequences derived from plastids and nuclei represent 1–12% and 0.1–13.4% of the mitogenome, respectively [71,72]. Most of these insertions occur in the intergenic regions, even though substitutions of protein-coding gene fractions have been cited [73]. One of the roles of chloroplast-derived sequences is to provide tRNA genes for mitochondrial translation [74], although these sequences may also function as promoters for mitochondrial genes [75]. Most of these chloroplast foreign sequences are non-coding or pseudogenic, although they sometimes appear in coding regions [76] due in particular to alterations in the genetic code between the two organelles [77] and/or differences in the initiation and termination signals of transcription and translation [78]. Remarkably, the amounts of DNA attributed to external sources (plastids and nuclei) are insufficient to account for the origins of all the intergenic regions because more than 60% of these regions show no detectable homology with other known sequences, and their conservation is too weak to support strong functional constraints [79].

### 1.5. Limits and Objectives of This Study

Another striking feature of angiosperm mitochondrial genomes is their low gene density and a very low collinearity [80,81]. Thus, although the size of their plant mitogenomes is variable, their conserved genetic pool is generally small, with sequences encoding central mitochondrial functions such as subunits of oxidative phosphorylation chain complexes (OXPHOS), proteins involved in the biogenesis of these complexes, and several ribosomal proteins [70,82]. Mitochondrial protein-coding genes typically undergo RNA modification processes to fulfill their functional roles through RNA editing, a post-transcriptional regulation mechanism [83,84].

Due to this remarkable plasticity, it is difficult to assemble plant mitochondrial genomes [85]. However, the recent hybrid assembly approach, which combines long and short reads has proven to be suitable [38,86,87].

In order to better understand the terrestrial adaptation mechanisms of *Lgh* and *Lpm*, and to augment the genomic reservoir of both species, we carried out de novo assemblies of the mitogenomes of *Lgh* and *Lpm*. To achieve this, we first generated *Lpm* and *Lgh* Illumina Mi-seq short-read (SR) sequences and/or Oxford Nanopore Technologies long reads (LR). Next, we used hybrid assemblies to combine the SR and LR sequences, facilitating the assembly of *Lgh* and *Lpm* mitogenomes. Afterward, we completed functional annotations and analyzed repeats and chloroplast insertions to determine the *Lpm* and *Lgh* mitogenome structures. Finally, we identified genes under selection and constructed a phylogenetic tree using mitogenomes from the Myrtales order.

## 2. Results

### 2.1. De Novo Assembly of Ludwigia Multichromosomal Mitogenomes

Sequencing of *Lgh* via Oxford Nanopore Technologies yielded a total of 482,619 reads (denoted as LRs for long reads), amounting to 2,386,362,939 bp. These reads exhibited an average length of 4944 bp, with the longest read extending to 72.9 kb. The sequencing conducted with Illumina Mi-seq technology resulted in 2 × 23,067,490 reads (abbreviated as SRs for short reads), totaling 9,606,411,104 bp. The reads had an average length of 208 base pairs, with the maximum read length reaching 301 base pairs. Error correction using Ratatosk for LRs and SPAdes for SRs retained 96% of the short reads and all long reads. Using these corrected reads, we conducted three distinct de novo assemblies using Megahit for SR-only assembly, Flye for LR-only assembly, and a hybrid assembly combining both read types using SPAdes.

Statistics for *Lgh* mitogenome assemblies were calculated using QUAST and are summarized in Appendix A. We were able to observe that the SPAdes assembly stood out for generating the largest contig (192 kb), the greatest total length (806 kb), and achieving the highest genome fraction (94.9%). However, it also exhibited the highest error rate, with 147 mismatches per 100 kb and three misassemblies. Conversely, Megahit exhibited the lowest error rate, tallying just 46 mismatches per 100 kb and zero misassembled contigs. However, it showed weakness in terms of completion, and only reconstructed 89% of the genome, which stood out as the lowest completion rate among the three assemblers. Finally, Flye emerged as the optimal choice, striking a balance between the two other assemblers, with no misassemblies and less mismatches (71 per 100 kb) than SPAdes, and better LGA50 and total aligned length (664 kb) than Megahit. Simultaneously, with these assemblies, we selected 1118 mitochondrial LRs using coding sequences (CDS) from the malvids clade species mitogenomes. These reads were assembled into 12 contigs with the Geneious assembler, allowing us to identify mitochondrial contigs from the de novo assemblies. Subsequently, we extended all of these contigs to achieve the complete mitogenome. Ultimately, we successfully assembled the *Lgh* mitogenome into two circular molecules, referred to as M1 and M2, with respective sizes of 544,782 bp and 166,796 bp, resulting in a combined total of 711,578 bp (Figure 1). The *Lgh* mitogenome coverage depths were 38x for LRs and 229x for SRs, and these values were similar for both M1 and M2. The *Lgh* mitogenome GC content was 45.2%.

Sequencing *Lpm* using Oxford Nanopore Technologies generated a dataset comprising 68,907 reads, which was reduced to 67,384 following self-correction by RATATOSK. These cumulatively amounted to 510,468,382 bp. These reads exhibited an average length of 7434 bp, with some extending as long as 83.03 kilobases. Without the availability of short reads, only the Flye assembly could be carried out and generated 496 contigs. Using the *Lgh* mitogenome as a reference, we found one *Lpm* contig of 158,656 kb that corresponded to 95% of the *Lgh* M2 molecule, and eight *Lpm* contigs that covered 83.02% of the *Lgh* M1 molecule (453,279 kb in total). Using these contigs and incorporating *Lpm* LRs that were not assembled by Flye, we successfully reconstructed the *Lpm* mitogenome, forming two circular molecules (also designated as M1 and M2), mirroring the structure seen in *Lgh*. These two molecules had respective sizes of 555,518 bp and 167,000 bp, culminating in a total mitogenome size of 722,518 bp (Figure 1). The coverage depth of the *Lpm* mitogenome was 10.1 for M1 and 5.8 for M2, with a GC content of 45.2%.

The mitogenomes of *Lgh* and *Lpm* were submitted to GenBank under the accession numbers PP727126, PP72717, PP727128, and PP727129.

### 2.2. Annotation and Comparisons of Ludwigia Mitogenome Contents

#### 2.2.1. Mitogenome Organization Comparisons

To assess the rearrangement and collinearity of the *Lgh* and *Lpm* mitogenomes, their mtDNA sequences were compared using ProgressiveMauve and Lastz (Figure 2). While the two M2 molecules were colinear and identical to 99.5%, the M1 counterparts were divided into eight locally collinear blocks (LCBs), collectively representing 88.43% and 90.27% of their total lengths. The sizes of these LCBs were well-preserved, with that of *Lpm* being consistently slightly larger than that of *Lgh* (Appendix A). The LCB positions varied, with (as shown in Figure 2C) three inverted blocks following a central symmetry (LCB3, LCB4, LCB7), highlighting extensive mitogenome rearrangements between the two species of *Ludwigia*. In the five inter-LCB regions larger than 500 bp, three contained a repeated sequence of 5253 bp that was conserved between the two *Ludwigia* species. Two of these contained the genes *rrn26* and *trnM-CAU*, and the third contained a region of approximately 6156 kb, also conserved between the two species, carrying the genes *rrn18* and *rrn5*, of which a copy was present at the end of LCB3.

#### 2.2.2. Protein Coding Gene Content

The mitogenomes of both *Ludwigia* species encompassed a total of 42–46 protein-coding genes, all located inside the LCBs, 68–70% of which were on the M1 molecule. Among these genes, 23 were involved in the oxidative phosphorylation (OXPHOS) machinery (Table 1), and of these, nine were associated with complex I (NADH dehydrogenase; *nad1*–*7*, *nad9*, and *nad4L*), with three trans-splicing genes (*nad1*, *nad2* and *nad5*) exhibiting distinct exon distributions. Exons of *nad2* were all localized on molecule M2, whereas exons of *nad5* were encoded on molecule M1, spread across LCB1 and LCB5, and exons of *nad1* were split between the two molecules. Specifically, exons 1, 4, and 5 resided in M1(LCB1 and LCB8), while exons 2 and 3 were located in M2. Significantly, the *matR* gene was encoded within exon 1 of *nad1*, a phenomenon consistently observed across all angiosperm mitogenomes (PMID: 24600456). The genes of the other complexes were as follows: *sdh4* for complex II (succinate dehydrogenase), *cob* for complex III (cytochrome c reductase; three genes in complex IV (cytochrome c oxidase; *cox1*–*3*), and five in complex V (ATP synthase; *atp1*, *4*, *6*, *8*, and *9*). Additionally, there were four subunits dedicated to the biogenesis of cytochrome c (*ccmB*, *ccmC*, *ccmFN*, and *ccmFC*), along with genes encoding maturase (*matR*) and transport membrane protein (*mttB*). These genes were distributed across the different LCBs. Furthermore, 14 non-core genes encoding small and large subunits of ribosome proteins were found to be present (Table 1). We identified seven cis-splicing protein-coding genes (*ccmFc*, *cox2*, *nad4*, *nad7*, *rpl2*, *rps3*, and *rps10*) and annotated seven protein-coding genes that originated from their respective plastids. The genes *petN* (encoding cytochrome b6f), *psbZ* (involved in photosystem II), and *rps14* (ribosomal protein S14) were transferred intact in both the *Lgh* and *Lpm* mitogenomes, whereas *ndhJ* and *ndhC* (NADH dehydrogenase), *atpF* (ATP synthase), and *rps2* (ribosomal protein S2) were intact in *Lpm* mtDNA but underwent pseudogenization in the *Lgh* mitogenome (Appendix A).

#### 2.2.3. RNA Editing

In the mitogenomes of *Lgh* and *Lpm*, 738 and 788 post-transcriptional C to U editing events, with a probability of more than 50%, were identified within the coding DNA sequences (CDS) (Appendix A). Among these, the *psab*, *nad4*, *ccmB*, and *mttB* genes exhibited the highest numbers of RNA editing sites, with 49, 46, 45, and 41 sites, respectively. Approximately 80% of the post-transcriptional modifications were non-synonymous (589–623 non-synonymous modifications compared to 149–165 synonymous modifications in *Lgh* and *Lpm*, respectively). Eight and twelve RNA editing events led to the creation of stop codons in the mitogenomes of *Lgh* and *Lpm*. These stop codon occurrences were predominantly observed in the chloroplast genes (*atpE*, *atpI*, and *psaB* in both mitogenomes, and *ndhK* in the *Lpm* mitogenome only), resulting in the annotation of these genes as pseudogenes. Additionally, two RNA editing sites generated stop codons in two mitochondrial genes (*atp9* and *ccmFc*), resulting in their truncation.

#### 2.2.4. Content of RNA Genes

In both *Ludwigia* mitogenomes, we identified and annotated a comprehensive set of 34 tRNA genes (Appendix A). This included 19 native mitochondrial tRNAs and an additional 15 tRNAs derived from the plastome. Among these, sixteen tRNA genes were unique whereas *trnM-CAU* was found in nine copies, *trnG-GCC* in three copies, and *trnN-GUU*, *trnP-UGG*, and *trnS-UGA* in two copies each. Three of them were cis-splicing tRNAs (*trnA-UGC*, *trnI-GAU*, and *trnV-UAV*), all of which originated from the plastome. As shown in Appendix A, all tRNAs were located in LCBs, except for the three *trnM-CAU* copies present in the previously described 5253 bp repeated region. As described previously, our analysis revealed the presence of multiple copies of mitochondrial ribosomal RNA genes (*rrn5*, *rrn18*, and *rrn26*; Appendix A) along with a singular copy of the plastid ribosomal RNA genes (*rrn4.5*, *rrn5*, and *rrn16*) (Appendix A).

#### 2.2.5. Chloroplast-Derived Fragments

In addition to the previously mentioned three to seven protein-coding genes, fifteen tRNA genes, and three rRNA genes, our analysis revealed twenty-four homologous fragments between the *Lgh* and *Lpm* chloroplast genomes (Figure 3; Appendix A). Cumulatively, these fragments had a size of approximately 55 kb, contributing to about 7.8% of the entire mitochondrial genome length in both *Ludwigia* species. Notably, these insertions of chloroplast DNA were dispersed throughout the entire M1 and M2 molecules (Figure 3), indicating potential independent transfers or regions modified through subsequent recombination. The largest of these fragments (8051 bp), located on the M2 molecule, was 98.4% identical to the original chloroplastic region present in both the *Lgh* and *Lpm* plastomes. This included the three pseudogenes *atpA*, *atpH*, and *atpI*, which harbors a stop codon due to post-transcriptional RNA editing, and the two genes *atpF* and *rps2*, which were complete in *Lpm* and pseudogenized in *Lgh* as well as a segment of the *rpoC2* gene. All pseudogenizations were associated with point mutations that led to frameshifts. The next largest fragment, spanning 7.6 kb and located on the M1 molecule, exhibited a lower identity to the plastome (93%) and featured an intact *trnM-CAU* gene at its extremity, accompanied by degraded forms of the *ycf2* and *ycf15* genes. In addition, we identified a 6.1 kb region with 99.6% identity to cpDNA, encompassing three tRNA genes, the *rrn16* gene, and a fragment of *rrn23*.

Another region, 5.4 kb in length, exhibited more degradation in *Lgh* compared to *Lpm* (89.3% and 91.9% identity with cpDNA, respectively), resulting in the pseudogenization of the *ndhJ* and *ndhC* genes in *Lgh* only, whereas *ndhK* and *atpE* were pseudogenized in both mitogenomes. Finally, the ultimate two extended regions of plastid origin, measuring 4.8 and 3.6 kb, respectively, were identical in both species and corresponded to the original cpDNA copy (Appendix A), albeit with gene fragments (*psaA*, *ycf1*, and *rrn23*) at their termini. The remaining plastidic-originated fragments, ranging in size from 1215 to 147 bp contained either complete tRNA sequences or, for the majority, fragments of protein-coding genes. Some of these corresponded to remains of *rps12*, *atpB*, or *psaA*, present in two copies (Appendix A).

#### 2.2.6. Fragments of Mobile Genetic Elements

Four main types of endogenous fragments of mobile genetic elements were widespread in both the M1 and M2 *Ludwigia* mitogenomes: mitovirus NERVEs (nonretroviral endogenized RNA virus elements), all three classes of nuclear retrotransposons, the *Tyl/copia*, *Ty3/gypsy*-, and non-LTR/LINE-families, and plasmid derivatives (Appendix A). For mitovirus NERVES, all fragments corresponded to partial or pseudogenized copies of the RNA dependent RNA polymerase (RdRP) of mitoviruses and were identical in *Lgh* and *Lpm*. The retrotransposon fragments integrated into the *Ludwigia* mitochondrial genomes were dispersed throughout the intergenic regions of the mitogenomes without showing a significant preference for integration hotspots. No complete copy was observed for any of the mobile elements identified.

#### 2.2.7. Repeated Sequences and Recombination

Three non-tandem repeats exceeding 1 kb in size were detected within both the *Ludwigia* M1 mitogenomes, whereas none were detected in the M2 molecules (Figure 4). The first repeat encompassed a central region spanning 5641 bp (called “Core R1”; Figure 4A), was found in triplicate within both *Ludwigia* M1 molecules, and carried a gene for both 26S rRNA (*rrn26*) and the initiator methionine tRNA *trnM-CAU*. This core region exhibited a 99.9% identity between *Lgh* and *Lpm*, except for a divergent trinucleotide at position 386 (TCC in *Lgh*, GGA in *Lpm*) and a dinucleotide at position 5261 (TC in *Lgh*, GA in *Lpm*). In the vicinity of this central “Core R1” area, we identified three scenarios: “R1-right” (Figure 4A), corresponding to an expansion in the 3′ region, which included both the “Extension” and “Right” regions present in *Lpm* and *Lgh* as well as an additional region called “Right_lgh” exclusive to *Lgh*; “R1-left” (Figure 4A), corresponding to an additional 5′ region (called “Left” for both and a small region unique to *Lgh* called “Left_lgh”), which was added to the “Core R1” part in *Lpm* and to the “Core_R1” and ”Extension” regions in *Lgh*; and finally, “R1-left-right” (Figure 4A), which combined both the “R1-right” and “R1-left” regions. The second and third repeats (R2 and R3) were classic repetitive sequences, each appearing twice in the *Lpm* and *Lgh* M1 molecules. The R2 repeats, which harbored a solitary gene encoding trnG-GCC at their 3′ termini, spanned 9384 and 9376 base pairs in *Lgh* and *Lpm*, respectively. These repeats showcased a remarkable 99.9% identity across the two *Ludwigia* mitogenomes. Notably, copies found in *Lgh* possessed an additional eight bases and exhibited two single nucleotide polymorphisms (SNPs) compared to those present in *Lpm*. R2 repeats were in direct orientation in *Lpm* but were reversed in *Lgh*. *Lpm* repeats R3 were 1554 nt longer than *Lgh* repeats R3, but their common regions of 7817 nt were highly conserved (99.9% identity, 5 SNPs). Like R2, the R3 repeats were in direct orientation in *Lpm* and reversed in *Lgh.* Placing these repeats within the framework of the locally collinear blocks (LCBs) shared between *Lpm* and *Lgh*, it was apparent that all R1 repeats and one of the R3 repeats were positioned in LCBs within non-conserved regions, as depicted in Figure 4B. Meanwhile, the second iteration of R3 resided within LCB3, close to a fragment of chloroplastic origin. Additionally, the two R2 repeats were identified on LCB1 and LCB5.

By using these large repeated sequences, multiple alternative conformations were tested. In the *Lgh* mitogenome, an alternative form of the M1 molecule was found with a probability of 50%, mediated by R1-right (Figure 4B and Figure 5). In this case, the M1 molecule was divided into two smaller molecules of 435 and 109 kb. Some other conformations mediated by R2, R1-core (The conserved part between R1-left and R1-right), and R3 repeats could not be completely validated but might also exist (Figure 4B and Figure 5). Recombinations mediated by these three repeats led to all molecules of the same size as the original M1 mitogenome, each with a specific inversion (Figure 5). By combining all of these recombinations, three molecules of 109, 174, and 261 kb could be obtained. In the *Lpm* mitogenome, an alternative form of the M1 molecule was identified as mediated by R1-right like in *Lgh* (Figure 6). This form was composed of two molecules of 425 and 130 kb. Due to the lack of reads, we could not identify as many putative forms as in the *Lgh* M1 molecule, and only one was found with R1-core (Figure 6). This consisted of an M1 molecule of 555 kb with an inversion. A combination of the two rearrangements may also exist.

A total of 266–294 non-tandem repeats (Appendix A) of at least 30 bp were identified in the *Ludwigia* mitogenomes and corresponded to 127–169 direct repeats and 125–139 reverse repeats. Non-tandem repeats represented 7–7.4% (50,752–52,800 bp) of each mitogenome. Most of them (161–185; 60.5–62.9%) were only found on M1, while 34-–36.8% (98–100) of repeats were found between the M1 and M2 molecules and 2.6–3% (7–9) were only on the M2 molecule. The majority of repeats (192–220; 72.2–74.8%) were between 30 and 59 bp (Appendix A). Only 10.2–13.5% (30–36) of repeats were larger than 100 bp. A total of 25 and 22 tandem repeats were found in the *Lgh* and *Lpm* mitogenomes (20–24 on M1 and one or two on M2; Appendix A). These repeats were in non-coding regions, except for the three of 18-period size present in each of the three copies of the rrn26 genes.

In the *Ludwigia* mitogenomes, 183–155 SSRs were identified (137–117 and 46–38 found on the M1 and M2 molecules, respectively; Appendix A). The majority of SSRs were monomeric (66–44) and tetrameric (63–56). Monomeric repeats were almost entirely composed of A or T (30–20 A and 34–24 T). A total of 32–36 dimeric, 15–13 trimeric, and 7–6 pentameric SSRs were identified. Dimeric repeats of TA were the third most common SSR (9–10) after monomeric A and T repeats.

### 2.3. Gene Comparisons and Phylogeny

Relative synonymous codon usage (RSCU) was calculated using MEGA in the *Lgh* and *Lpm* mitogenome CDS on 11,595 and 12,141 codons, respectively (Figure 7; Appendix A). A total of 30–29 codons had an RSCU > 1, which means that they were used more often than another synonymous codon. Among them, the most preferentially used codons were A-ended or U-ended, except for the UGG leucine codon, showing a strong bias in the third letter of the codon. In both mitogenomes, the UGA stop codon had the biggest RSCU value (1.6–1.64). In *Lgh*, the smallest RSCU value was for the UAG stop codon (0.29) while in *Lpm*, it was the GCG alanine codon (0.42).

To identify genes under selection, Ka (non-synonymous substitution rate) and Ks (synonymous substitution rate) were calculated on 31 genes by comparing *Ludwigia* mitogenomes (these genes were identical in *Lgh* and *Lpm* mitogenomes) to each of the other 14 assembled Myrtales mitogenomes one by one, and to the *Geranium maderense* mitogenome as an outgroup. Genes under positive selection (not conserved) will have a Ka/Ks value above one, whereas genes under negative selection (conserved) will have a Ka/Ks value under one. A Ka/Ks ratio value of one points to a neutral selection. The *rps1* gene is the only gene showing a Ka/Ks >1 in the majority of species studied (Figure 8, Appendix A). Moreover, the *ccmB* and *mttB* genes were also under negative selection in *Lgh* and *Lpm* in seven and five species, respectively. However, the majority of Ka/Ks ratios (92%) were under 1, showing the conservation of most genes.

Sixteen conserved protein-coding genes were concatenated and aligned to make a phylogenetic tree of the Myrtales order with *Geranium maderense* used as an outgroup (Figure 9). The tree fit with the phylogeny in the Myrtales order. The outgroup *G. maderense* was separated from the Myrtales order species. Species from the same family were grouped. The four Onagroideae subfamily species were clustered together within the Onagraceae family. Finally, the two *Ludwigia* species were on the same branch.

Gene compositions were compared in the 15 Myrtales order species and *G*. *maderense* mitogenomes (Figure 10). Sixteen genes were common between all of the studied mitogenomes. Ten genes were only missing in one mitogenome. Among them, nine were missing in *R. tomentosa* and one in *E. urophylla* × *grandis*. Mitochondrial genes missing in more than one mitogenome were ribosomal subunit genes (*rpl16*, *rps7*, *rps10*, *rps12*, *rps13*, and *rps19*), *ccmFn* and *sdh3*. Out of all species, 83% of the identified chloroplast protein coding genes were incomplete. Only five species contained one to nine complete chloroplast genes: *Lgh*, *Lpm*, *M*. *candidum*, *M*. *sanguineum*, and *E*. *grandis*. These genes coded for different functions: PSI assembly factor (*ycf3*), photosystem II (*psbM* and *psbZ*), ATP synthase (*atpF*, *atpH*, and *atpI*), cytochrome b6f (*petN*), NADH dehydrogenase (*ndhB*, *ndhC*, and *ndhJ*), transcription (*rpoC1*), ribosomal large subunit (*rpl14*, *rpl22*, and *rpl23*), and ribosomal small subunit (*rps2* and *rps14*). Each of these genes were present in only one or two species, always of the same genus.

## 3. Discussion

In this study, *Lgh* and *Lpm* mitogenomes were assembled de novo from the SR or/and LR sequences. For *Lgh*, the best assembling combination was found to be SPAdes and Flye. Two circular molecules called M1 and M2 were obtained for both mitogenomes and had similar sizes (544,782 bp and 555,518 bp for M1 and 166,796 bp and 158,656 bp for *Lgh* and *Lpm*, respectively). For these two closely related species, their mtDNAs showed similarities in their structures and sequences for both the M1 and M2 molecules. While the M2 molecules were highly similar (99.5%) and colinear, the M1 molecules from *Lpm* and *Lgh* mitogenomes contained eight conserved blocks including three inverted blocks.

### 3.1. Comparison of Plant Mitogenomes in Myrtales Order

Plant mitogenomes vary greatly in size, ranging from 66 kb to 12 Mb. In the Myrtales order, which comprises 9 families, 399 genera, and about 13,000 species [88], only the mitochondria genomes of 17 species have been sequenced. Most of these sequenced mitogenomes are around 400 to 450 kb in size, as follows for the Lythraceae family *(Punica granatum* (404,807 bp) [89], *Lagerstroemia indica* (333,948 bp; NC 035616), *Trapa incisa* (381,774 bp; NC 086691), *T. bicornis* (383,262 bp; NC 086690)); Myrtaceae family (*Rhodomyrtus tomentosa* (400,482 bp) [90], *Eucalyptus grandis* (478,813 bp) [91], *E. camaldulensis* (463,134 bp) [92], *E*. *urophylla* × *grandis* (481,982 bp; OQ947725-OQ947727), *Syzygium samarangense* (530,242 bp; NC 079700)); Melastomataceae family (*Medinilla magnifica* (377,864 bp) [93], *Melastoma candidum* (391,595 bp; NC 071383), *M*. *sanguineum* (395,542 bp; NC 071384), and *M*. *dodecandrum* (411,954 bp) [94]); and the Onagraceae family *(Oenothera villaricae*, *O. biennis* and *O. elata*, with respective lengths of 408,260, 424,132, and 418,451 bp [37]) and *Chamaenerion angustifolium* (495,176 bp; OX328283)). For *Lgh* and *Lpm*, however, the total length of mitogenomes was notably higher, standing at approximately 700 kb (711,578 bp and 722,518 bp, respectively), and even the largest M1 molecules alone exhibited a greater size than that of the mitogenomes of the previous 17 species.

The origin of this variation in size between different plant mitochondrial genomes is multifaceted, encompassing the proliferation of repeat elements, the integration of foreign DNA, and the acquisition or loss of large intragenic segments [95]. Homologous recombination has led to rearrangements that play a significant role in plant mitogenome evolution [96]. In their study, Zhou et al. (2023) conducted pairwise comparisons of the mitogenomes of five Myrtales species and observed sequence similarities ranging from 17% to 36% [94]. Notably, the highest percentage of shared sequences was observed between *M. magnifica* and *M. dodecandrum*, which belong to the same family. Similarly, the comparison of the mitogenomes of *E. camaldulensis* and *E. grandis* revealed a distinct alteration in repeat sequences, shedding light on evolutionary changes between the two genomes [92]. In the mitogenomes of *Lgh* and *Lpm*, the smaller M2 molecules exhibited colinearity and shared 99.5% identity, whereas the larger M1 molecules displayed rearrangements in five inter-LCB regions (size > 1 kb) between the two species, although no alterations were observed in the repeat sequences. One explanation for this result could be the genomic proximity of these closely related species, a hypothesis supported by the fact that *Lpm* is one of the maternal progenitor species of *Lgh* [33]. Given that most plants inherit mitochondrial genomes from the maternal parent, the minimal difference observed between the two mitogenomes might signify the evolutionary divergence between *Lpm* and *Lgh*.

### 3.2. Existence of Multi-Chromosomal Mitogenomes

Initially, most plant mitogenomes were depicted as single circular molecules. However, the increased availability of plant mitogenomes has unveiled a greater complexity, with multi-chromosomal mitogenomes identified in many species. However, research by Wu et al. (2020) indicates that multiple chromosome structures are present in only about 10% of sequenced plant mitogenomes [95]. Significantly, the mitochondrial genomes of *Lgh* and *Lpm* stand out as inaugural examples of multiple mitogenome structures documented within the Myrtales order. In this context, Bi et al. (2020) posed an intriguing question regarding the mechanisms that regulate and control the replication and segregation of multi-chromosomal mitogenomes during cell division [97]. In our study, we observed that for *Ludwigia*, sequence coverage was consistent for both the M1 and M2 molecules, indicating an equal number of copies of both molecules in the cell and therefore the likelihood of equi-replication.

Recently, Yang et al. (2022) reconstructed the mitogenome of *Salvia miltiorrhiza*, which was initially assembled as a single circular molecule through short-read sequence analysis [98]. By employing both long-read and short-read technologies, the researchers identified two mitochondrial chromosomes, highlighting the value of combining both sequencing technologies. This finding reinforces our research, demonstrating that the integration of short-read (SR) and long-read (LR) sequences leads to improved mitogenome assembly. Notably, SR sequences, which are shorter than the repeat size, may not fully span the repeats independently, highlighting the necessity of combining both technologies for comprehensive assembly.

The presence of large repeat sequences (longer than 1 kb) has been shown to induce structural changes or isomerization in plant mitogenomes [58,99]. Hence, recent studies suggest the presence of one to two dominant forms of sub-genomic mitochondrial chromosomes, with repeat sequences capable of mediating recombination, leading to a diverse array of minor conformations. For instance, in *Gelsemium elegans*, four pairs of repeat sequences have been shown to facilitate the formation of one major and five minor conformations of the mitochondrial genome [100]. In *Salvia miltiorrhiza*, Yang et al. (2022) discovered two mitochondrial genomes and identified multiple minor conformations resulting from homologous recombination mediated by nine repeat sequences [98]. In our study, we identified two dominant forms, M1 and M2, and validated, through long-read (LR) sequence mapping, the existence of two minor conformations for the M1 molecule in both *Lgh* and *Lpm*, mediated by three repeat sequences. Other configurations are theoretically possible, but could not be confirmed with the sequences available.

### 3.3. Plastid DNA Transfer in Lpm and Lgh Mitogenomes

Inter-organelle DNA transfer can occur between the mitochondria and plastids, although it is extremely rare in the direction of mitogenomes to plastids. The plastome sequences represent 0% to 10% of the mitochondrial genome. The presence of chloroplast sequences and the number of genes in mitochondrial genome are very fluctuant. Recently, Wang et al. (2018) demonstrated that plastid-derived DNA fragments comprised 10.34% of the entire mitogenome length of *P. micranthus*, encompassing twelve intact protein-coding genes, three tRNAs, and twenty-nine pseudogenes [101]. Similarly, Tang et al. (2024) observed for *Acorus tatarinowii* and Yang et al. (2024) for *Haloxylon Ammodendron* that homologous sequences between the mitochondrial genome and the chloroplast genome accounted for 3.78% and 2.49%, respectively [102,103]. In our study, both the *Lgh* and *Lpm* M1 and M2 molecules contained plastome-origin sequences, encompassing 21–25 intact genes and constituting 7.8% of the mitochondrial genome length, which is quite a high percentage compared to the studies cited above, particularly given the large size of *Ludwigia* mitogenomes.

RNA editing is a post-transcriptional mechanism in higher plant organelles, and contributes to improved protein folding [104]. The conversion of C-to-U is the main form of RNA editing, as we observed in *Lgh* and *Lpm* [105]. RNA editing plays a crucial role in various plant developmental processes and could serve as a vital mechanism in adapting to a changing environment, allowing plants to acclimatize and survive [106]. Given the remarkable plasticity of both *Ludwigia* species to thrive in aquatic to dry environments [23], the role of RNA editing in facilitating this acclimatization warrants investigation. We identified the highest number of RNA editing sites among all mitochondrial genes in the *psab*, *nad4*, *ccmB*, and *mttB* genes. Similarly, Lu et al. (2024) reported elevated RNA editing sites in the *ccmB* and *nad4* genes in *Syzygium samarangense* [107]. Additionally, Xiong et al. (2017) investigated the responses of six mitochondrial genes including *ccmB* to oxidative stress, commonly associated with abiotic stresses like drought [108]. They demonstrated that differences in the RNA editing of *ccmB* could modulate plant performance responses under oxidative stress.

To summarize, the *Lpm* and *Lgh* mitogenomes consisted of two circular molecules with similar structures and sequences. The presence of large repeat sequences could result in multiple minor conformations. Plastome-origin sequences accounted for 7.8% of both mitochondrial genomes. A high number of RNA editing was observed.

## 4. Materials and Methods

### 4.1. Plant Material and Sequencing Data

To avoid genetic variability, *Lgh* and *Lpm* plants used for the sequencing were obtained using vegetative reproduction from a single fragment of stem with buds for each species. DNA extractions were carried out from new buds according to a specific protocol for extracting DNA long fragments [32]. DNA fragments were sequenced using Illumina MiSeq for short reads (SRs) and Oxford Nanopore GridION for long reads (LRs) at the Genome Transcriptome Facility of Bordeaux (INRAE-UMR 1202 BIOGECO, Bordeaux, France), as described in Barloy-Hubler et al. (2023). Both LRs and SRs were available for *Lgh* but only LRs for *Lpm*. Quality controls and filtering were performed using Guppy v4.0.14 for the nanopore reads and fastp v0.20.0 for the MiSeq reads [109]. Short reads were then paired and merged using BBMerge [110].

### 4.2. Assembly Strategies and Annotations

For the assembly of the *Lgh* mitogenome, SRs were corrected with LRs using SPAdes and LRs with SRs using Ratatosk [111,112]. Subsequently, Flye was used to assemble the corrected LRs, Megahit to assemble the corrected SRs, and SPAdes for the corrected SR-LR hybrid assemblies [113,114].

To recruit mitochondrial reads, we developed a strategy centered on the coding DNA sequences (CDSs) due to their high conservation across species [57]. Therefore, all mitogenomes from the malvids clade were retrieved from the NCBI RefSeq database, and each CDS was identified, aligned, and a consensus sequence was constructed to serve as references to map corrected LRs. LRs selected in this manner were then assembled using the ‘De Novo Assemble’ tool within Geneious Prime 2023 (http://www.geneious.com/, accessed on 28 June 2024) and the resulting contigs were elongated by successive mapping of the LRs, SRs, and SPAdes (only for *Lgh*) and Flye assemblies. The *Lpm* mitogenome was assembled both de novo and using the *Lgh* mitogenome as a reference. For *Lgh*, as we had both LRs and SRs, we compared the hybrid (SPAdes), LR-only (Flye), and SR-only (Megahit) assemblies using the QUality ASsessment Tool (QUAST) [115].

CDS and tRNA annotations were performed using Geseq and BLASTx [116,117]. Rfam was used to annotate the rRNA [118]. All annotations were manually verified and biocurated. OGDraw was used for the graphic representation of the mitogenomes [119]. For the RNA editing predictions, CDSs were extracted and Deepred-MT was used to find the C-to-U post-transcriptional modifications [120]. Five genes were only analyzed in *Lpm* (*atpF*, *ndhC*, *ndhJ*, *ndhK*, and *rps2*) as they were annotated as pseudogenes in the *Lgh* mitogenome. Mobile elements were annotated using DANTE via the RepeatExplorer Galaxy server and each region identified was checked for both gene designation and boundaries using BlastN and BlastP [121].

### 4.3. Sequence Analysis

REPuter on BiBiServ2 (https://bibiserv.cebitec.uni-bielefeld.de, accessed on 28 June 2024) identified direct and reverse repeats (maximum computed repeats = 500; minimal repeat size = 8) [122]. Simple sequence repeats (SSR) were annotated by MISA with the parameters set to 10, 5, 4, 3, 3, and 3 for the mono-, di-, tri-, tetra-, penta-, and hexa-nucleotides, respectively [123]. Tandem Repeats Finder was used to identify tandem repeats [124]. Chloroplast sequence insertions in the mitochondrial genome were found using ProgressiveMauve with the plastome [125]. Circular repeats maps were made using shinyCircos [126]. Each repeat longer than 1 kb was extracted, along with 1 kb regions on each side, and recombination events were constructed. Then, the LRs were mapped to all of these sequences using the ‘Map to Reference’ function in Geneious, and sequences that overlapped the repeats and covered at least 100 bp of the two flanking 1 kb regions were tallied for all conformations and utilized to identify potential recombinations. If the repeats were in the same direction, the molecule was divided into two molecules at the repeat locations. If the repeats were in the opposite direction, an inversion occurred between the two repeats. A general sequences comparison of the *Ludwigia* mitogenomes was conducted by using LastZ [127]. ProgressiveMauve identified LCBs (locally collinear blocks) between the *Lgh* and *Lpm* mitogenomes.

### 4.4. Codon Usage Bias and Ka/Ks Calculation

All CDSs were extracted from each *Ludwigia* mitogenome and concatenate. MEGA was used to count the presence of each codon and calculate their RSCU (relative synonymous codon usage) [126].

The complete mitogenomes of 13 species belonging to the Myrtales order were collected from the NCBI RefSeq database: *Oenothera biennis* (MZ934756), *Oenothera elata* (MZ934757), *Oenothera villaricae* (MZ934755) [37], *Chamaenerion angustifolium* (OX328283), *Lagerstroemia indica* (NC 035616), *Eucalyptus grandis* (NC 040010) [91], *Eucalyptus urophylla* × *grandis* (OQ947727), *Melastoma dodecandrum* (OR148386) [94], *Melastoma candidum* (NC 071383), *Melastoma sanguineum* (NC 071384), *Punica granatum* (NC 071229) [89], *Rhodomyrtus tomentosa* (NC 071968) [90], and *Syzygium samarangense* (NC 079700) [107]. In addition to these 13 mitogenomes, the mitogenome of *Geranium maderense* (NC 027000) from the Geraniales order was also collected and used as the outgroup species. *C. angustifolium* was annotated with Geseq [116]. Thirty-one genes present in at least 14 of the 16 studied mitogenomes (14 collected in database and *Lpm* and *Lgh*) were selected (*nad1*, *nad2*, *nad3*, *nad4*, *nad4L*, *nad5*, *nad6*, *nad7*, *nad9*, *sdh4*, *cob*, *cox1*, *cox2*, *cox3*, *atp1*, *atp4*, *atp6*, *atp8*, *atp9*, *ccmB*, *ccmC*, *ccmFc*, *matR*, *mttb*, *rpl2*, *rpl5*, *rpl10*, *rps1*, *rps3*, *rps4*, and *rps14*). Protein coding genes were aligned using MAFFT, and MEGA was used to calculate Ka (non-synonymous substitution rate) and Ks (synonymous substitution rate) with the Nei–Gojobori (Jukes–Cantor) method, of these 31 genes between *Lgh* as the reference, and the other 15 species [127,128].

### 4.5. Molecular Mass, Isoelectric Point and Subcellular Localization Predictions

Sequences of mitochondrial protein coding genes were translated by Geneious. Protein sequences were used to calculate the molecular mass and isoelectric point with Sequence Manipulation Suite [129]. These were also used for subcellular localization prediction with Plant-mPLoc [130].

### 4.6. Phylogenetic and Comparative Analysis

Mitogenomes used in the Ka/Ks calculation were also used for phylogenetic analysis. Sixteen genes (*nad1*, *nad2*, *nad4*, *nad5*, *nad6*, *nad9*, *cob*, *cox1*, *cox3*, *atp4*, *atp6*, *atp8*, *ccmB*, *ccmC*, *ccmFc*, and *rpl10*), common between the 16 mitogenomes selected, were concatenated and aligned with MAFFT. MEGA was used to make a tree with the maximum likelihood method using a bootstrap of 1000 r.

## 5. Conclusions

To conclude, we obtained the first high-quality mitogenomes of two *Ludwigia* species, *Ludwigia peploides* subsp. *montevidensis* and *Ludwigia grandiflora* subsp. *hexapetala*. The *Lpm* and *Lgh* mitogenomes were the first of the Ludwigioideae subfamily to be sequenced, which regroups 83 species, and the fifth and sixth in the Onagraceae family, which includes approximately 660 species, and correspond to the eighteenth and nineteenth to be sequenced in the Myrtales order. The fact that there are so few mitogenomes in the Myrtales order compared to plastomes (342 in the RefSeq database in April 2024) highlights how challenging it is to assemble mitogenomes. In addition, *Lgh* and *Lpm* mitogenomes were the first to be found as two circular molecules in the Myrtales order. Our results can be used as a reference for the enrichment of mitochondrial resources in the Myrtales order, which is considered to be one of the largest in the Angiosperms with around 12,000 species [131].

## Figures and Tables

**Figure 1 ijms-25-07283-f001:**
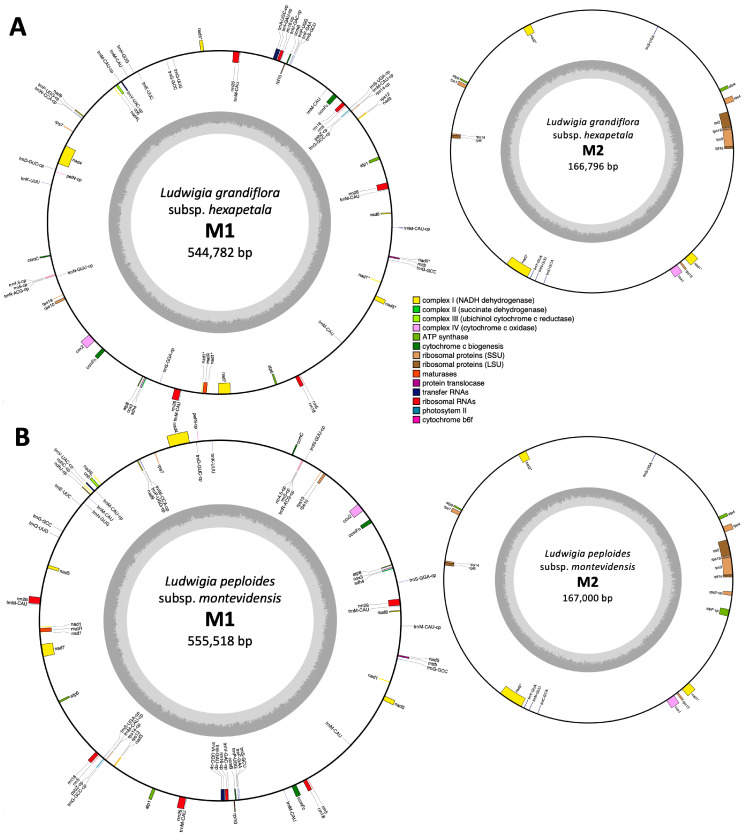
The two circular mitochondrial genomes of *Ludwigia grandiflora* subsp. *hexapetala* (**A**) and *Ludwigia peploides* subsp. *montevidensis* (**B**). Genes outside the circle are transcribed in a counter-clockwise direction, while inner genes are transcribed clockwise. Spliced genes are marked with an asterisk *. Genes labeled with -cp originate from the chloroplast. Inner gray circles represent the GC content. Genes are color-coded, as indicated in the legend.

**Figure 2 ijms-25-07283-f002:**
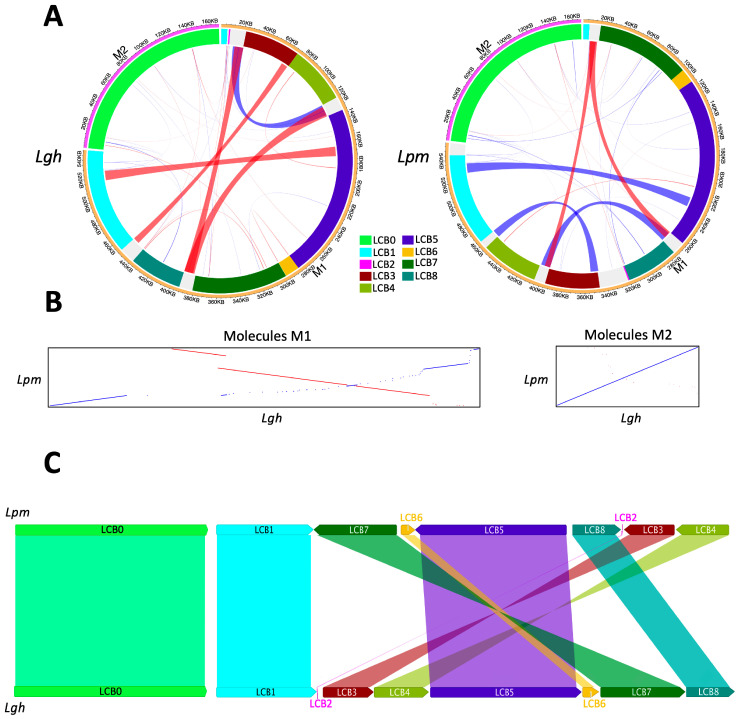
Structural comparisons between the *Ludwigia grandiflora* subsp. *hexapetala* (*Lgh*) and *Ludwigia peploides* subsp. *montevidensis* (*Lpm*) mitogenomes. (**A**) Repeats spanning at least 100 bp are illustrated with curved lines, with the width of the lines proportional to the size of the repeats. Inverted repeats are denoted in red, while direct repeats are indicated in blue. LCBs (locally collinear blocks) represent conserved regions between the *Lgh* and *Lpm* mitogenomes, as depicted in panel (**C**). (**B**) Comparisons of the M1 and M2 sequences of the two *Ludwigia* species were performed using LastZ: blue lines indicate common sequences in direct orientation, while red lines denote sequences that are common but in inverted orientation. (**C**) Locally collinear blocks (LCBs) (>1 kb) identified by ProgressiveMauve between the mitogenomes of *Lgh* and *Lpm*.

**Figure 3 ijms-25-07283-f003:**
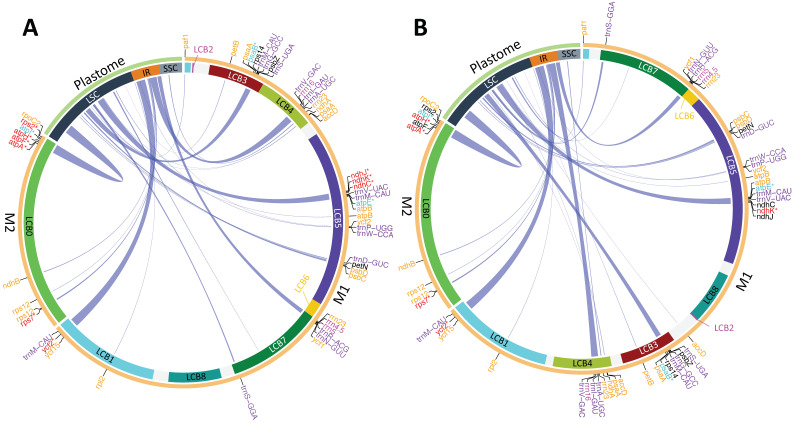
Maps illustrating the chloroplast sequence insertions (>80 bp) in the mitogenomes *Ludwigia grandiflora* subsp. *hexapetala* (*Lgh*; (**A**)) and *Ludwigia peploides* subsp. *montevidensis* (*Lpm*; (**B**)). Curved lines connect chloroplast insertions in the mitogenome to their respective origins in the plastome and the line width is proportional to the insertion size. Locally collinear blocks (LCBs) represent conserved regions between the *Lgh* and *Lpm* mitogenomes, identified by ProgressiveMauve. Plastome regions including large single copy (LSC), inverted repeat (IR; represented once for simplification), and small single copy (SSC), are indicated. Genes that originated from the plastome are color-coded: pseudogenes in red, pseudogenes due to RNA editing in blue, fragment genes in orange, complete protein coding genes in black, tRNA genes in purple, and rRNA genes in magenta. * Correspond to pseudogene.

**Figure 4 ijms-25-07283-f004:**
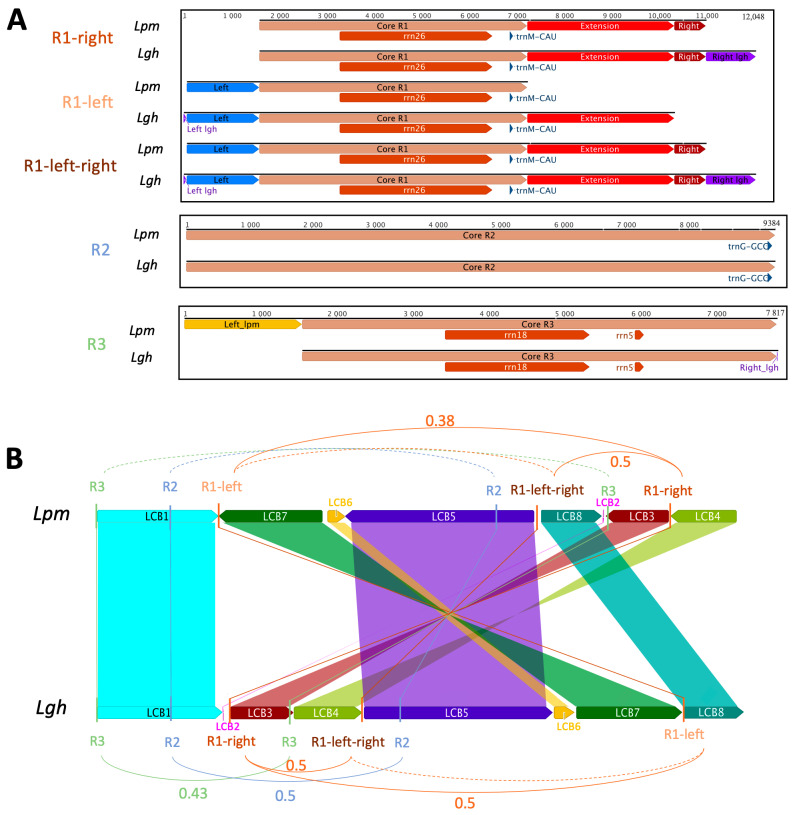
This representation of repeats exceeding 1 kb in the mitogenomes of *Ludwigia grandiflora* subsp. *hexapetala* (*Lgh*) and *Ludwigia peploides* subsp. *montevidensis* (*Lpm*). “R” denotes repeat. R1-left-right combines R1-right and R1-left. (**A**) The five repeats exceeding 1 kb in the mitogenomes of *Lpm* and *Lgh*. Core sequences represent conserved sequences among repeats. “Extension” sequences were present in all repeats except R1-left in *Lpm*. Left and right sequences were unique to R1-left and R1-right, respectively, and both were present in R1-left-right. Purple and yellow sequences were unique to *Lgh* and *Lpm*, respectively. (**B**) Sequence comparison of M1 molecules between the two *Ludwigia* species and their repeats. Locally collinear blocks (LCBs) represent conserved regions between the *Lgh* and *Lpm* mitogenomes, identified by ProgressiveMauve. Recombination frequencies are specified when >0, while a dotted line indicates the absence of recombination.

**Figure 5 ijms-25-07283-f005:**
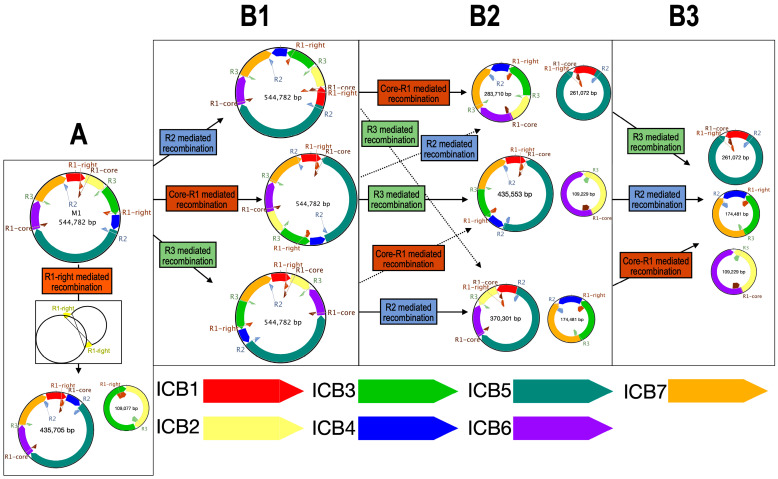
Schematic of mitogenome *Lgh* recombinations: (**A**) Validated conformations after M1 molecule rearrangement in *Ludwigia grandiflora* subsp. *hexapetala* mitogenome. (**B1**–**B3**) Different steps of the feasible successive recombinations. Only intramolecular rearrangements are depicted. “R” denotes repeat. ICB (intramolecular collinear block) represents regions conserved among all forms.

**Figure 6 ijms-25-07283-f006:**
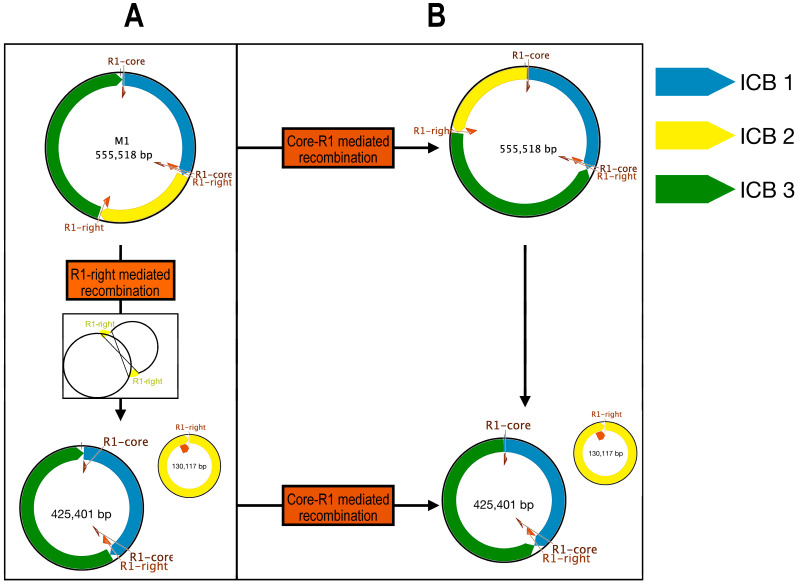
Schematic of mitogenome *Lpm* recombinations: (**A**) Validated conformations after M1 molecule rearrangements of *Ludwigia peploides* subsp. *montevidensis* mitogenome. (**B**) Putative conformations after M1 molecule rearrangements of *Ludwigia peploides* subsp. *montevidensis* mitogenome. Only intramolecular rearrangements are depicted. “R” denotes repeat. ICB (intramolecular collinear block) represents regions conserved among all forms.

**Figure 7 ijms-25-07283-f007:**
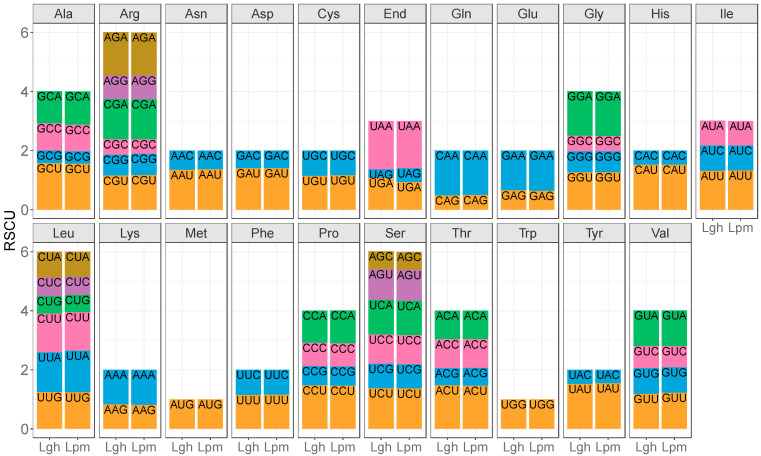
The relative synonymous codon usage (RSCU) bias in *Ludwigia grandiflora* subsp. *hexapetala* (*Lgh*) and *Ludwigia peploides* subsp. *montevidensis* (*Lpm*) mitogenome CDS (sequences of protein coding genes).

**Figure 8 ijms-25-07283-f008:**
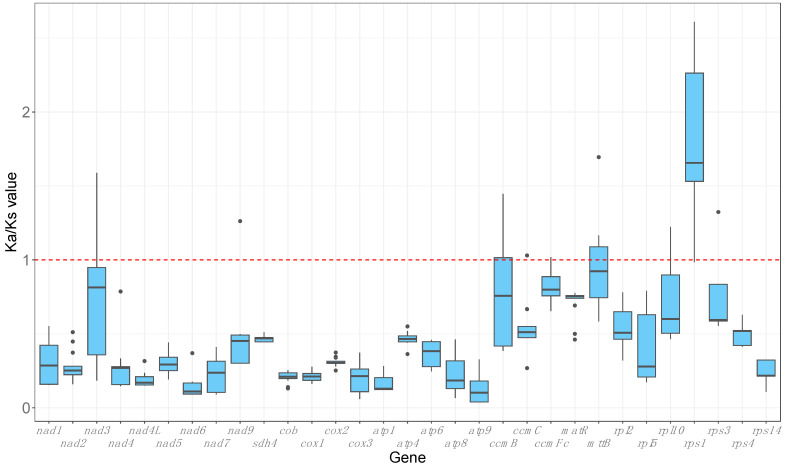
Graphical representation of pairwise comparisons of non-synonymous substitution rate (Ka) and synonymous substitution rate (Ks) ratios among species from the orders Myrtales and Geraniales, utilizing *Ludwigia grandiflora* subsp. *hexapetala* and *Ludwigia peploides* subsp. *montevidensis* as references, across 31 mitochondrial genes. Genes were included if they were present in at least 14 out of the 16 selected species.

**Figure 9 ijms-25-07283-f009:**
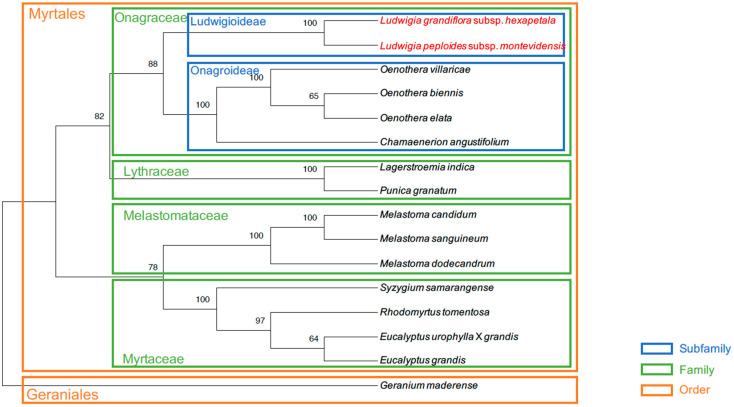
Phylogenetic tree of the Myrtales order constructed using 16 protein-coding genes via the maximum likelihood method with a bootstrap value of 1000. *Geranium maderense* was designated as the outgroup. Branch frequencies are denoted by numbers on branches. The *Ludwigia* species sequenced in this paper are highlighted in red.

**Figure 10 ijms-25-07283-f010:**
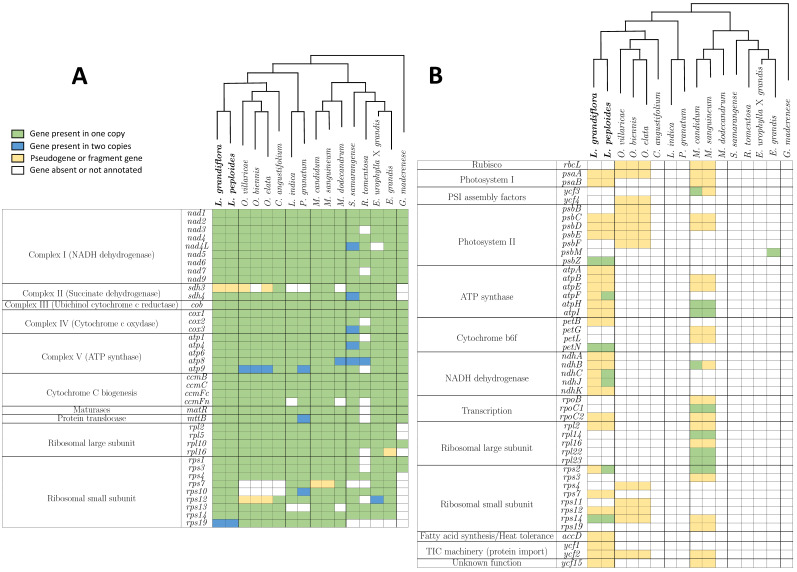
Gene compositions of mitogenomes from 15 Myrtales and one Geraniales order species. Panel (**A**) displays the mitochondrial genes, while Panel (**B**) shows the chloroplast genes. The phylogenetic trees presented were based on 16 protein-coding genes and constructed using the maximum likelihood method (refer to Figure 9). The *Ludwigia* species sequenced in this paper are highlighted in bold. Color boxes correspond to presence/absence panel (see legend in figure).

**Table 1 ijms-25-07283-t001:** Mitochondrial protein coding genes annotated in the *Ludwigia grandiflora* subsp. *hexapetala* (*Lgh*) and *Ludwigia peploides* subsp. *montevidensis* (*Lpm*) mitogenomes. Numbers within parentheses indicate the number of exons for spliced genes. Trans-spliced genes are marked by “*”. Locally collinear blocks (LCBs) correspond to conserved regions between *Lgh* and *Lpm* mitogenomes, identified by ProgressiveMauve. The molecular weight and isoelectric point of protein were calculated using the Sequence Manipulation Suite. Hypothetical localization was estimated by Plant-mPLoc.

Gene	LCB	Molecule	Size (in bp)	Molecular Weight (in kDa)	Protein Isoelectric Point	Hypothetical Localization
Complex I (NADH dehydrogenase)						
*nad1* (5*)	LCB0	M2	1008	36.81	10.10	Mitochondrion
	LCB1	M1				
	LCB8	M1				
*nad2* (5*)	LCB0	M2	1518	55.34	9.28	Mitochondrion
*nad3*	LCB3	M1	357	13.39	4.33	Mitochondrion
*nad4* (4)	LCB5	M1	1473	54.27	8.97	Mitochondrion
*nad4L*	LCB5	M1	303	10.94	6.70	Mitochondrion
*nad5* (5*)	LCB1	M1	1794	74.55	7.90	Mitochondrion
	LCB5	M1				
*nad6*	LCB1	M1	618	23.45	10.53	Mitochondrion
*nad7* (5)	LCB8	M1	1182	44.32	7.87	Mitochondrion
*nad9*	LCB5	M1	573	22.63	8.20	Mitochondrion
Complex II (succinate dehydrogenase)						
*sdh4*	LCB7	M1	393	15.25	10.76	Mitochondrion
Complex III (cytochrome c reductase)						
*cob*	LCB5	M1	1182	44.04	7.52	Cell membrane
Complex IV (cytochrome c oxidase)						
*cox1*	LCB0	M2	1623	58.75	9.25	Mitochondrion
*cox2* (2)	LCB7	M1	780	29.09	5.53	Mitochondrion
*cox3*	LCB7	M1	798	29.54	7.53	Mitochondrion
Complex V (ATP synthase)						
*atp1*	LCB3	M1	1533	55.51	6.21	Mitochondrion
*atp4*	LCB0	M2	573	21.25	10.26	Mitochondrion
*atp6*	LCB8	M1	1044	38.30	4.80	Mitochondrion
*atp8*	LCB7	M1	480	18.19	9.50	Mitochondrion
*atp9*	LCB0	M2	330	11.61	9.09	Mitochondrion
Cytochrome c biogenesis						
*ccmB*	LCB4	M1	621	23.07	8.62	Mitochondrion
*ccmC*	LCB6	M1	723	26.92	10.98	Mitochondrion
*ccmFc* (2)	LCB4	M1	1500	56.40	10.74	Mitochondrion
*ccmFn*	LCB7	M1	1983	74.21	10.45	Mitochondrion
Maturase						
*matR*	LCB8	M1	1983	73.76	10.71	Chloroplast
Protein translocase						
*mttb*	LCB1	M1	768	29.07	7.75	Mitochondrion
Ribosomal large subunit						
*rpl2* (3)	LCB0	M2	1389	50.69	11.28	Mitochondrion
*rpl5*	LCB0	M2	564	21.55	5.89	Mitochondrion
*rpl10*	LCB4	M1	486	18.53	7.45	Cell membrane, chloroplast, and nucleus
*rpl16*	LCB0	M2	435	15.98	11.81	Mitochondrion
Ribosomal small subunit						
*rps1*	LCB0	M2	693	25.87	11.89	Mitochondrion
*rps3* (2)	LCB0	M2	1659	63.57	10.96	Mitochondrion
*rps4*	LCB0	M2	1047	41.18	11.63	Mitochondrion
*rps7*	LCB5	M1	447	17.04	11.21	Chloroplast and mitochondrion
*rps10* (2)	LCB7	M1	408	16.40	11.25	Mitochondrion
*rps12*	LCB3	M1	378	14.30	11.80	Chloroplast and mitochondrion
*rps13*	LCB0	M2	351	13.27	11.21	Mitochondrion
*rps14*	LCB0	M2	303	11.96	12.04	Mitochondrion
*rps19*	LCB0	M2	285	11.20	12.32	Chloroplast
	LCB7	M1				

## Data Availability

The original contribution presented in the study are included in this article (see Appendix A). Accession numbers of the *Lpm* and *Lgh* mitogenomes are indicated in the main manuscript.

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
