# Peer review of "De Novo Hybrid Assembly Unveils Multi-Chromosomal Mitochondrial Genomes in Ludwigia Species, Highlighting Genomic Recombination, Gene Transfer, and RNA Editing Events"

_ijms, 2024, doi:10.3390/ijms25137283_

Round 1

Reviewer 1 Report

Comments and Suggestions for Authors

Dear Editor,

The present manuscript by Doré et al. provides the first complete sequencing and analysis of the mitochondrial genomes of Lgh and Lpm, along with a detailed phylogenetic analysis. I would like to accept this article for possible publication in the International Journal of Molecular sciences after following the minor suggestions:

Q1. Figure Legends

In each figure legend, ensure the scientific names are italicized. Additionally, to maintain consistency throughout the manuscript, the abbreviations "Lgh" and "Lpm" should also be italicized.

Q2. Numbers

To maintain consistency, numbers less than 10 should be written in words (one to nine), while numbers 10 and greater should be written as numerals.

Q3. Page 5 - Line 201

"figure" should be corrected to "Figure".

Q4. Page 12 - Lines 388-403

There are various instances where the sequences should be in ascending order, but they are currently in reverse. Please check and correct the order of the sequences:

Line 390: 7-7.4% (50,752-52,800 bp)

Line 391: 34-36.8% (98-100)

Line 393: 192-220 (72.2-74.8%)

Line 394: 10.2-13.5% (30-36)

Line 395: 20-24

Q5. Page 15 - Line 470

Check the figure "1666,796 bp"; it seems an extra “6” might have been added. Verify and correct this mistake and look for similar errors throughout the manuscript.

Regards,

Author Response

Q1. Figure Legends: In each figure legend, ensure the scientific names are italicized. Additionally, to maintain consistency throughout the manuscript, the abbreviations "Lgh" and "Lpm" should also be italicized.

Action: We updated each legend accordingly and reviewed the changes within the text.

Q2. Numbers: To maintain consistency, numbers less than 10 should be written in words (one to nine), while numbers 10 and greater should be written as numerals.

Action: We made the changes throughout the entire manuscript.

Q3. Page 5 - Line 201: "figure" should be corrected to "Figure".

 Action: correction done

Q4. Page 12 - Lines 388-403: There are various instances where the sequences should be in ascending order, but they are currently in reverse. Please check and correct the order of the sequences:

Line 390: 7-7.4% (50,752-52,800 bp)

Line 391: 34-36.8% (98-100)

Line 393: 192-220 (72.2-74.8%)

Line 394: 10.2-13.5% (30-36)

Line 395: 20-24

Action: corrections done

Q5. Page 15 - Line 470: Check the figure "1666,796 bp"; it seems an extra “6” might have been added. Verify and correct this mistake and look for similar errors throughout the manuscript.

Action: correction and verification done

Reviewer 2 Report

Comments and Suggestions for Authors

Dear Authors,

Your manuscript titled „De novo hybrid assembly unveils multi-chromosomal mitochondrial genomes in Ludwigia species, highlighting genomic recombination, gene transfer, and RNA editing events” contains very intersting results. Due to fact, that it reffers to invasive plant species it might interest an international audience. However, I have found some imperfections, which-in my opinion- should be improved or at least clarified before an eventual publication. I have listed them below:

1.       Lines 36-40. I suggest to enlarge description of studies species, especially with information about individual traits leading to succesful colonization of new areas (e.g. lifespan; reproduction mode: generative and vegetative; seed production and their dispersal).

2.       Lines 132-138. In my opinion the chapter Introduction should ended by listing of specific aims of investigations.

3.       I encourage Authors to take into consideration the dividing of large chapters Intoduction and Discuussion into some separate  subchapters.

4.       Figures 1-10 in current form are illegible mainly due to too small dimensions of fonts.

5.       The way of citing the literature sources in chapter References should be improved according the guide for Authors.

Author Response

1- Lines 36-40. I suggest to enlarge description of studies species, especially with information about individual traits leading to successful colonization of new areas (e.g. lifespan; reproduction mode: generative and vegetative; seed production and their dispersal).

Action: To address this comment, we have added the following paragraph (lines 42-51):

“Lpm and Lgh show some advantageous biological traits which might explain their success of colonization in news areas. Lpm and Lghreproduce essentially by clonal propagation with high grow rate, which contribute to propagules dispersal and establishment [6]. Both Ludwigia species are capable of thriving in a wide range of environments, revealing their broad ecological tolerance and great plasticity [7]. Sexual production is also effective in both Ludwigia species. In France, Lpm is self-compatible and produces many capsules and seeds [8]. Lgh, on the other hand, possess a heteromorphic late-acting self-incompatible system and also able to produce seeds (Portillo-Lemus et al, 2021). Seeds can be dispersed by floating fruit, which contributes to long-distance colonization, as is the case for propagules [7].”

2 - Lines 132-138. In my opinion the chapter Introduction should ended by listing of specific aims of investigations.

Action: In response to this suggestion, we have revised the paragraph as follows (lines 152-158):

“To achieve that, we first generated Lpm and Lgh Illumina Mi-seq short reads (SR) sequences and/or Oxford Nanopore technologies long reads (LR). Next, we used hybrid assemblies to combine SR and LR sequences, facilitating the assembly Lgh and Lpm mitogenomes. Afterward, we completed functional annotations and analyzed repeats and chloroplast insertions to determine Lpm and Lgh mitogenome structures. Finally, we identified genes under selection and constructed a phylogenetic tree using mitogenomes from the Myrtales order.”

3 - I encourage Authors to take into consideration the dividing of large chapters Introduction and Discussion into some separate  subchapters.

Action: We have followed this suggestion and divided the introduction and discussion into sub-chapters. These changes are indicated in the text with titles highlighted in yellow.

4 - Figures 1-10 in current form are illegible mainly due to too small dimensions of fonts.

Action: All figures have been changed. See text.

5 - The way of citing the literature sources in chapter References should be improved according the guide for Authors.

Action: We have revised the format of all references to align with the authors' guidelines.

Reviewer 3 Report

Comments and Suggestions for Authors

The paper is a complete description on an uncharacterized topic, as is mithocondrial genomes from invasive species. The research is well conducted, , the results are sound and the manuscript is well written.

I only have two formal criticisms.

In Figures 5,6 and 10 the lettering is too small. I would enlarge them and use a bigger font in order to increase the readibility.

I would also add a conclusion at the end of the discussion section summarizing the major traits of the newly described genomes.

As there are only about of 40 coding genes i would include a table describing them, with their major traits (kd, Pi, hypothetical localization) in the text, not as supplementary.

Author Response

1 - In Figures 5,6 and 10 the lettering is too small. I would enlarge them and use a bigger font in order to increase the readibility.

Action: These corrections have been addressed as required.

2 - I would also add a conclusion at the end of the discussion section summarizing the major traits of the newly described genomes.

Action: As requested, we have added this summary paragraph to the end of the discussion section (lines 638-641) :

“To summarize, Lpm and Lgh mitogenomes consisted of two circular molecules with similar structures and sequences. The presence of large repeat sequences could result in multiple minor conformations. Plastome-origin sequences accounted for 7.8% of both mitochondrial genomes. A high number of RNA editing were observed.”

3 - As there are only about of 40 coding genes i would include a table describing them, with their major traits (kd, Pi, hypothetical localization) in the text, not as supplementary.

Action: Following this suggestion, we have reintegrated this table into the main text (now Table 1) and included information on the molecular mass, isoelectric point, and predicted subcellular localization for each protein. The corresponding paragraph detailing these predictions has also been added in the Materials and Methods section (lines 716-720).